# LDINet: Latent Decomposition and Interpolation for Single Image FMO Deblatting

## Abstract

The image of fast-moving objects usually contains a blur stripe indicating the blurred object that is mixed with backgrounds. To deblur the stripe and separate the object from the background in this single image, in this work we propose a novel LDINet that introduces an efficient decomposition-interpolation module (DIB) to generate the appearances and shapes of the objects. In particular, under the assumption that motion blur is an accumulation of the appearance of the object over exposure time, in the latent space the feature maps of the long blur is decomposed into several shorter blur parts. Specifically, the blurry input is first encoded into latent feature maps. Then the DIB module breaks down the feature maps into discrete time indexed parts corresponding to different small blurs and further interpolates the target latent frames in accordance with the provided time indices. In addition, the feature maps are categorized into the scalar-like and gradient-like classes which help the affine transformations effectively capture the motion of feature warping in the interpolation. Finally, the sharp and clear images are rendered with a decoder. Extensive experiments are conducted and has shown that the proposed LDINet achieves superior performances compared to the existing competing methods.

## 1 Introduction

Motion deblurring is a special case of the deblurring task that aims to restore a clear and sharp image or images from a blurred one caused by the moving of the object and/or the camera. Conventional methods (Kupyn et al., 2018; 2019; Wieschollek et al., 2017; Sim & Kim, 2019) for motion deblurring mostly recover a clear image at the median of the motion. Recently, some works (Jin et al., 2018; Purohit et al., 2019; Xu et al., 2021; Zhong et al., 2022; Rozumnyi et al., 2021a; Zhong et al., 2023) further focus on the finer structures of the blur and learn to generate sequences of clear images of the object in chronological order, which is known as sequence from blur or single image temporal super-resolution task.

In this work, we focus on a special case of the sequence from blur task, i.e., deblatting of fast-moving objects (FMOs). FMOs, first defined by Rozumnyi et al. (2017), are moving objects that move over a distance greater than their size within the exposure time of the camera in the scene. As a result, the blurry portion of an FMO becomes a stripe due to the long-distance moving, which makes it hard to distinguish the object's appearance. The deblatting task aims to accomplish two goals, i.e., image deblurring which generates a sequence of clear and sharp images from the blurry input, and image matting which separates the object in the scene from the background. Additionally, in this task, it is assumed that the camera is fixed with static backgrounds, which is accordance with many real-world scenarios. For example, in sports analysis, we want to detect and track the fast-moving balls in pictures and videos which are captured by static cameras.

Formally, given an input pair of pictures, including one background picture and one picture with the blurred fast-moving object, our goal is to generate a sequence of sharp appearances and masks for deblurring and matting according to the given time indices. Since the time indices could be chosen arbitrarily from a continuous interval, the targeting result is considered as a mapping to sharp appearances and mask with a variable $t$ which indicates the given time index. From the physical perspective, the formation process of the motion blur can be formulated as the integral of the mapping over the exposure time. Since solving the integral is difficult, several approximations

have been proposed to simplify the formation model. Kotera & Šroubek (2018) and Kotera et al. (2019) approximate the blurring and matting formation model by convolving a fixed moving blur kernel with the appearance and shape of the object. However, a simple convolution kernel cannot capture the variance of a moving object. To this end, TbD-3D (Rozumnyi et al., 2020) approximates the integral with a piecewise linear model and leverages energy minimization to solve the deblatting task. Specifically, the blurring stripe is considered as the sum of several small blurs according to a partition of the exposure time and each small blur is approximated by a linear motion blur. However, the linear approximation is performed in the image space directly and cannot well capture motions with rotation when the shapes of the FMOs are complex. Furthermore, the inference time consumption caused by the energy minimization method is usually prohibitively expensive.

On the other hand, DeFMO (Rozumnyi et al., 2021a) have firstly proposed to solve the deblatting task with an encoder-decoder structure in a data-driven way. With a large-scale training dataset and elaborately designed loss terms, DeFMO outperforms previous methods in terms of both the performance and inference time cost. However, it adopts shared latent embedding for different time index and directly concatenates the specific index with the embedding as intermediate features for decoding, which would limit the flexibility of generating diverse outputs for different time indexes.

To address the limitations mentioned above, we propose our LDINet to exploit the intrinsic structure of the latent space and naturally introduce the time index via an interpolation architecture. In particular, a simple yet effective decomposition-interpolation structure is introduced in the neural network model for FMO deblatting, where the feature maps in latent space output by the encoder are decomposed into several latent parts according to a fine partition of the exposure time. For any time index in the exposure time interval, an interpolation method with affine transformations is proposed to aggregate the adjacent parts into a latent frame for further decoding with a bi-branched decoder to predict the mask and appearance on the time index. Further, since the complex mapping of the encoder would introduce nonlinear behaviors, the application of the affine transformation is not that direct as in the traditional image space. In light of that the convolution operation could be regarded as a linear combination of summation and series of directional derivatives which are linear projections of the gradient fields, the latent part of feature maps is disentangled into two categories, the scalar fields and the gradient fields. And the affine transformations are applied to the fields in two ways according to their categories.

Extensive comparison experiments and ablation studies are conducted to show the effectiveness of our designs for FMO deblatting and our LDINet also achieves competitive performance compared to the existing competing methods.

## 2 DEBLATTING METHOD

In this section, we first introduce the task setting of FMO deblatting and give an overview of the proposed LDINet. Then the well designed Decomposition-Interpolation Block (DIB) in our LDINet is elaborated in Section 2.1. Finally the corresponding learning objectives are presented in Section 2.2.

**Preliminary**. Given the appearance $F_t$ and the mask $M_t$ of the moving object at any time $t$ within the exposure time which is rescaled to $[0, 1]$, the resultant blurred FMO image $I$ could be formulated as

$$I = \int_0^1 F_t \, M_t + (1 - M_t)B \, \mathrm{d}t, \tag{1}$$

where $B$ is the static background. However, in the deblatting task for FMO, based on a blurred image $I$ and an estimated background $B$, the goal is to approximate a sharp rendering $R_\tau = [F_\tau \, M_\tau]$ at a given time index $\tau \in [0, 1]$. Here both of the image $I : D \to \mathbb{R}^3$ and background $B : D \to \mathbb{R}^3$ are RGB images where $D \subset \mathbb{R}^2$ is the canvas. The rendering $R_\tau : D \to \mathbb{R}^4$ is an RGBA image where the RGB part is the appearance $F_\tau$ and the alpha part is the mask $M_\tau$. Our estimation for the rendering $R_\tau$ is denoted as $\widehat{R}_\tau = [\widehat{F}_\tau \; \widehat{M}_\tau]$. Besides, during training, the renderings $\{R_{\tau_i}\}_{i=1}^n$ of equally spaced time indices $\{\tau_i\}_{i=1}^n$ are available in the dataset, where $\tau_i = \frac{i-1}{n-1}$.

**The deblatting pipeline**. As shown in Figure 1, our LDINet is composed of an encoder, a DIB module, and a decoder. In particular, the encoder first takes as input a blurred image $I$ and a background image $B$ and outputs feature maps $V$. Then a Decomposition-Interpolation Block (DIB) is introduced to decompose $V$ and interpolate the target latent frame $Q_\tau$ for the given time index

Figure 1: **Deblatting pipeline and the structure of the decoder.** The encoder first encodes the input pair $I, B$ into feature maps $V$. Then the DIB module synthesizes the latent frame $Q_\tau$ at time index $\tau$ from $V$. Finally, the bi-branched decoder generates the rendering $R_\tau$ which is consisting of the mask $M_\tau$ and the appearance $F_\tau$ from $Q_\tau$.

$\tau$, which would be further explained in the following section. Finally, the decoder generates the rendering $R_\tau$ with the target latent frame $Q_\tau$. The decoder is composed of several shared layers and two branches, which estimate the mask $M_\tau$ and appearance $F_\tau$ separately.

## 2.1 THE DECOMPOSITION-INTERPOLATION BLOCK

In this section, we elaborate the proposed Decomposition-Interpolation Block (DIB), which aims to explore the structure of the latent space more appropriately to generate a better latent frame for the target time index.

Compared with the conventional deblurring tasks, the main differences of the FMO deblurring tasks are the longer blurred stripe and more complex motion trajectory of the object, which makes it difficult to be resolved. However, if we consider a small time segment $\Delta t$ of the total exposure time interval $\Delta T$, the size of the blurred stripe within this time segment is small and the motion of the object is much simpler, which can be approximated by a linear motion as in Rozumnyi et al. (2020). From this point of view, the blur formation model in Equation 1 can be reformulated as

$$I = \sum_{k=0}^{m-1} \int_{k\Delta t}^{(k+1)\Delta t} F_\tau M_\tau + (1 - M_\tau)B\mathrm{d}\tau \approx \frac{1}{m}\sum_k H_{t_k} * F_{t_k} + (1 - H_{t_k} * M_{t_k})B, \quad (2)$$

where $\Delta t = \frac{1}{m}$ and $H_{t_k}$ is the kernel containing the motion information around the time index $t_k = \frac{k-1}{m-1}\}$.

Inspired by this observation, we consider that the feature maps could also be decomposed into a set of parts corresponding to a series of discrete time indices. Then the latent frame of the target time index would be obtained by interpolation. In particular, the feature maps $V$ are first decomposed into $m$ discrete latent parts $\{P_{t_i}\}_{i=1}^m$ in the latent space corresponding to the $m$ time indices $\{t_i\}_{i=1}^m$ with a projector. Here we assume that the $i$-th part $P_{t_i}$ contains the motion and appearance information of the object around the time index $t_i$. Then given the target time index $\tau$, the required latent frame $Q_\tau$ could be synthesized with $\{P_{t_i}\}_{i=1}^m$ by interpolation.

Though interpolation can be accomplished with a simple affine transformation in the traditional image space, the operation is more complex in the latent feature space. Specifically, since the FMOs we deal with are mostly rigid objects, the changes of the appearances of moving object that are adjacent in time indices can be modeled by affine transformations in the original image space. However, the complex mapping of the encoder would introduce nonlinear behaviors for the affine transformations in the feature space. In our case where the encoder is a convolution network, for a single input channel, the convolution operation could be regarded as a linear combination of summation and series of directional derivatives. Since the directional derivatives are linear projections of the gradient fields, it is reasonable to represent the convolution results by scalar fields and gradient fields. Although the scalar fields of the target frame can still be obtained by affine transformations from those of the neighboring parts, the gradient fields of the target frame could not be obtained in the same way.

To be specific, according to our analysis, the latent part $P_t$ is composed of a scalar field $P_t'$ and a gradient field $P_t''$. On the one hand, for the scalar field, the transformation behavior is the same as the affine transformation in the image space,

$$P_\tau'(\mathbf{x}) = P_t'(A(\mathbf{x})) \quad (3)$$

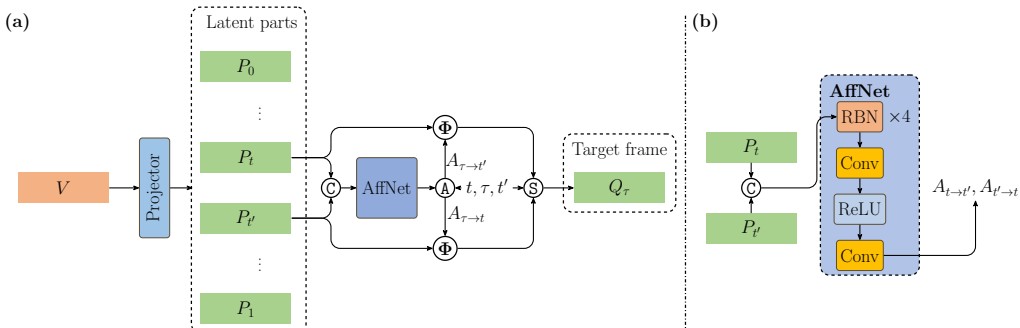

Figure 2: **The decomposition-interpolation block (DIB)**. (a) The overview of the decomposition-interpolation block. (b) The detail structure of the AffNet.

where $A$ is the affine transformation describing the motion from time index $\tau$ to $t$ on the point $\mathbf{x}$ in 2D coordinate. On the other hand, for the gradient field, under the assumption that it is the gradient of scalar field $S_t$, i.e., $P''_t(\mathbf{x}) = \partial_{\mathbf{x}} S_t(\mathbf{x})$, the behavior of the transformation on the gradient field becomes

$$P''_\tau(\mathbf{x}) = \partial_{\mathbf{x}} S_\tau(\mathbf{x}) = \partial_{\mathbf{x}'} S_t(\mathbf{x}') \frac{\partial \mathbf{x}'}{\partial \mathbf{x}} = P''_t(A(\mathbf{x})) \tilde{A}, \tag{4}$$

where $\mathbf{x}'$ is the transformed version of $\mathbf{x}$ under the affine transformation $A$ and $\tilde{A}$ is the Jacobian of $A$. Therefore, it is necessary to interpolate the scalar field and gradient field of the target time index in different ways. In particular, we first suppose that each latent part $P_t$ could be represented by a concatenation of the scalar field $P'_t$ and the gradient field $P''_t$, i.e., $P_t = [P'_t\ P''_t]$, and thus they would be processed separately. Then we denote $\mathbf{\Phi}[A, P]$ as a function which applies the affine transformation $A$ to the latent part $P$ in a way that the scalar field $P'_t$ and the gradient field $P''_t$ are first transformed by Equation 3 and Equation 4 respectively, and then concatenated as the result. In this way, the latent frame of the target time index would be approximated more appropriately.

Based on the above transformation method, one remaining difficulty is how to estimate the affine transformation in the feature space. To this end, we introduce a residual network named AffNet, as shown in 2 (b), which takes as input two latent parts and predicts a pair of affine transformations between them. Since there are several downsampling layers in the encoder, the size of the feature maps shrink several times. Thus each grid in the feature maps indeed contains the information of a patch of the input image. Therefore, we predict the affine transformations point-wisely that the AffNet generates affine transformations for each grid of the latent parts separately. Further, to make sure that the AffNet is capable of predicting the affine transformation between two parts as expected, a pretraining stage is also introduced to train the AffNet in advance by enforcing it to predict a randomly generated small affine transformation.

Finally, given the affine transformations estimated by AffNet, our interpolation process is presented more formally. In particular, as shown in Figure 2, to interpolate the latent frame $Q_\tau$ at the time index $\tau \in [0, 1]$, we first find the two nearest latent parts $P_t$ and $P_{t'}$ from the decomposition results, which satisfy $t \leq \tau \leq t'$, to obtain the affine transformations $A_{t \to t'}$ and $A_{t' \to t}$ between these two parts by AffNet. Next, to obtain the affine transformations from time $\tau$ to $t$ and from time $\tau$ to $t'$, we approximate them by $A_{\tau \to t} = I + \frac{\tau - t}{t' - t}(A_{t' \to t} - I)$ and $A_{\tau \to t'} = I + \frac{t' - \tau}{t' - t}(A_{t \to t'} - I)$, respectively. Then the target latent frame is interpolated with the affine transformations as

$$Q_\tau = \frac{t' - \tau}{t' - t} \mathbf{\Phi}[A_{\tau \to t}, P_t] + \frac{\tau - t}{t' - t} \mathbf{\Phi}[A_{\tau \to t'}, P_{t'}].$$

Note that here a weighting scheme is employed to fully leverage the information from both the two neighboring parts. Besides the application of $\mathbf{\Phi}$ is along the grids of the latent parts with the corresponding affine transformations point-wisely.

## 2.2 TRAINING LOSS

In this section, we introduce the training objectives of our LDINet, which can be divided into two categories according to the space where the constraints are performed, i.e., the image and the latent space. In particular, in the image space, a reconstruction loss $\mathcal{L}_R$ is introduced to reconstruct the masks, the appearances, and the blurry input, and a sharpness loss $\mathcal{L}_S$ is employed to sharpen the masks. As for the latent space, three objectives $\mathcal{L}_L$, $\mathcal{L}_{id}$, and $\mathcal{L}_C$ are introduced to encourage the feature invariance to different background, stabilize the training of the AffNet, and improve the feature consistency between adjacent latent parts, respectively.

**Direction of motion trajectory**. Before introducing the details of the loss functions, we first clarify the correspondence between the predicted sequence $\{\widehat{R}_{\tau_i}\}_{i=1}^n$ and the ground truth $\{R_{\tau_i}\}_{i=1}^n$. Specifically, since the motion blur keeps invariant even when the motion trajectory is reversed, the direction of the motion trajectory is ambiguous in fact. In order to determine the direction of the motion trajectory, we use the relative error rate of the masks

$$Err(\widehat{R}, R) = \sum_{\tau} \frac{\sum_D |\widehat{M}_\tau - M_\tau|}{\sum_D M_\tau}, \tag{5}$$

as the criteria and select the direction with a smaller relative error rate. Here, $\sum_D$ is the an operator that sums over the pixels in canvas $D$. For simplicity, with some abuse of notation, $\{\widehat{R}_{\tau_i}\}_{i=1}^n$ is used in the following description to represent the estimated rendering sequence in the selected direction.

**Reconstruction loss**. The reconstruction of the renderings at a given time index consists of three parts, i.e., the reconstruction of the mask, the appearance, and the blurry input. We use Binary Cross Entropy (BCE) loss for the reconstruction of the mask and L1 loss for the reconstruction of the appearance and the input. In particular, for the reconstruction of the appearance, the constraint is performed between the estimated and ground truth instance images $\widehat{I}_\tau = \widehat{M}_\tau \widehat{F}_\tau + (1 - \widehat{M}_\tau)B$ and $I_\tau = M_\tau F_\tau + (1 - M_\tau)B$ instead of between the estimated and ground truth appearances $\widehat{F}_\tau$ and $F_\tau$. As for the reconstruction of the blurry input, it encourages the consistency between the rendering model and the formation model of the blurry input in a self-supervised manner, where the estimation of the blurry input $\widehat{I} = \frac{1}{n}\sum_\tau \widehat{I}_\tau$ is enforced to match the blur input image. Besides, a shape-aware weighting scheme $W_\tau$ is further presented to reweight the appearance loss of each pixel based on its location to the outline of the object. In practice, the weighting scheme is obtained by blurring the mask $M_\tau$ with an average kernel $K_{avg}$, i.e., $W_\tau = K_{avg} * M_\tau$. Thus the overall reconstruction loss $\mathcal{L}_R$ is

$$\mathcal{L}_R = \frac{1}{n}\sum_\tau \frac{1}{\sum_D M_\tau}\sum_D \left(\ell_{BCE}(\widehat{M}_\tau, M_\tau) + W_\tau \ell_1(\widehat{I}_\tau, I_\tau)\right) + \frac{1}{\sum_D[(\sum_\tau M_\tau) > 0]}\sum_D \ell_1(\hat{I}, I), \tag{6}$$

where $\ell_{BCE}$ is the point-wise BCE loss, $\ell_1$ is the point-wise L1 loss, and $\sum_D[(\sum_\tau M_\tau) > 0]$ denotes the number of pixels within the FMO blur.

**Mask sharpening loss**. To sharpen the predicted mask, we propose to further strengthen the correct prediction results in the estimated mask, by decreasing the prediction entropy for the correctly classified pixels

$$\mathcal{L}_S = \frac{1}{n|D|}\sum_\tau \sum_D \mathbb{H}(\widehat{M}_\tau G_\tau), \tag{7}$$

where $G_\tau$ is a one-hot map that indicates the correctly classified pixels and $\mathbb{H}$ is the point-wise binary entropy.

**Background reduction loss**. Considering that the rendering results should be invariant to the background change, $\mathcal{L}_L$ is designed to reduce the influence of background on the feature maps. Specifically, given two inputs $X$ and $X'$ which only differ in the background, they are first encoded as $V$ and $V'$ in the latent space and then constrained by

$$\mathcal{L}_L = \ell_{MSE}(V, V'), \tag{8}$$

where $\ell_{MSE}$ is the mean square error loss.

**Feature consistency between latent parts**. Since the motion trajectory of the object are continuous, we consider that the latent frames should also be similar when the corresponding time indices are

close. Since the latent frames are the interpolated from the latent parts, we formulate the feature consistency loss $\mathcal{L}_C$ as

$$\mathcal{L}_C = \frac{1}{m-1} \sum_{i=1}^{m-1} \ell_{MSE}(P_{t_i}, P_{t_{i+1}}).$$ (9)

**Reversibility of the affine transformations**. Since the output of the AffNet is a pair of forward and backward affine transformations between the two input latent parts, we intend to constrain the two affine transformations to be the inverse of each other,

$$\mathcal{L}_{id} = \ell_{MSE}(A_{t \to t'} A_{t' \to t}, \mathbf{I}),$$ (10)

where $I$ is the identity matrix and $A_{t \to t'}$ and $A_{t' \to t}$ are the forward and backward transformations between the two latent parts at the time indices $t$ and $t'$ estimated by AffNet, respectively.

**Joint loss.** Consequently, the joint loss function is a combination of the two aspects,

$$\mathcal{L}_{joint} = \underbrace{\mathcal{L}_R + \mathcal{L}_S}_{\text{image space}} + \underbrace{\mathcal{L}_{id} + \mathcal{L}_L + \alpha_C \mathcal{L}_C}_{\text{latent space}}.$$ (11)

## 3 TRAINING AND EVALUATION

In this section, the training and evaluation datasets are first introduced in Section 3.1 and 3.2 and the training details are provided in Section 3.3. Then the proposed LDINet is compared with the existing state-of-the-art methods in Section 3.4. Finally, extensive ablation studies are conducted to evaluate the effect of each component in LDINet in Section 3.5 and visualization results are provided in Section 3.6.

### 3.1 SYNTHESIZED TRAINING DATASET

The synthesized dataset for training is based on the one from DeFMO (Rozumnyi et al., 2021a), which is generated with Blender Cycles (Community, 2018). Each training sample is created by a 3D object moving through a 6D linear trajectory over two background sequences and consists of two backgrounds for background reduction, one FMO blur stripe for the construction of blurry inputs, and 25 discrete frames of sharp renderings of the object at equally spaced time indices within the exposure time $[0, 1]$ including the start and end time. The 3D objects are sampled from ShapeNet (Chang et al., 2015) dataset applied with DTD (Cimpoi et al., 2014) textures. The backgrounds for training are sampled from the VOT (Kristan et al., 2016) sequences, and the backgrounds for validation are sampled from Sports-1M (Karpathy et al., 2014).

There are 50,000 samples for training and 1,000 samples for validation.

### 3.2 EVALUATION DATASET

The evaluation datasets are three real-world datasets from the FMO deblatting benchmark (Rozumnyi et al., 2021a):

**The TbD (Kotera et al., 2019)** is composed of 12 sports sequences with uniformly colored and mostly spherical objects. Each sequence contains 16~60 frames.

**The TbD-3D (Rozumnyi et al., 2020)** is composed of 10 sequences and contains objects with complex textures, and thus it is more difficult. Each sequence contains 37~81 frames. The rotations of the objects result in significant differences in their appearances. One limitation is that the objects are mostly spherical, so their shapes remain constant when rotated.

**The Falling Objects (Kotera et al., 2020)** is composed of 6 sequences and is the most challenging benchmark with objects of complex textures and 3D shapes. Each sequence contains 11~19 frames.

For each dataset, the low-speed sequences are created by averaging over the full exposure high-speed ground truths. The ground truths have a frame rate that is 8 times higher than that of the low-speed sequences.

Table 1: **Evaluation and Comparing results** on Falling Objects (Kotera et al., 2020), TbD-3D (Rozumnyi et al., 2020) and TbD (Kotera et al., 2019) datasets. The compared methods are TbD(Kotera et al., 2019), TbD-3D(Rozumnyi et al., 2020), BiT++(Zhong et al., 2023), and DeFMO(Rozumnyi et al., 2021a).

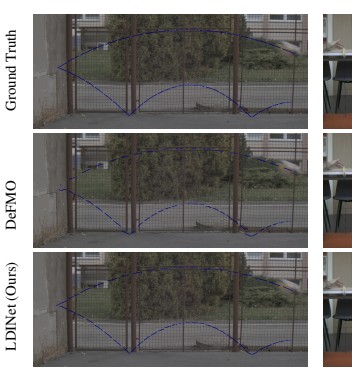

Figure 3: Estimation of trajectories on sequences from the TbD-3D dataset (Rozumnyi et al., 2020) (left) and Falling Objects dataset (Kotera et al., 2020) (right)

| Dataset | Score | Compared Methods | | | | Proposed |
|---|---|---|---|---|---|---|
| | | TbD | TbD-3D | BiT++ | DeFMO | LDINet |
| Falling | TIoU↑ | 0.539 | 0.539 | N/A | 0.684 | **0.692** |
| | PSNR↑ | 20.53 | 23.42 | 23.92 | 26.83 | **28.09** |
| | SSIM↑ | 0.591 | 0.671 | 0.596 | 0.753 | **0.770** |
| TbD-3D | TIoU↑ | 0.598 | 0.598 | N/A | 0.879 | **0.905** |
| | PSNR↑ | 18.84 | 23.13 | 24.98 | 26.23 | **26.56** |
| | SSIM↑ | 0.504 | 0.651 | 0.605 | 0.699 | **0.706** |
| TbD | TIoU↑ | 0.541 | 0.542 | N/A | 0.550 | **0.611** |
| | PSNR↑ | 23.22 | 25.21 | 24.69 | **25.57** | 25.312 |
| | SSIM↑ | 0.605 | **0.674** | 0.502 | 0.602 | 0.617 |

## 3.3 TRAINING SETTINGS

The training contains two stages, the pretraining stage and the finetuning stage. In the pretraining stage, we aim to train the AffNet and provide a guidance to disentangle the original latent part into the scalar part and gradient part. In particular, since the DIB module is not well trained at beginning, we use the interpolation method with weighted summation over the latent parts during the pretraining. To train the AffNet, we first introduce a pseudo input by applying a randomly generated small affine transformation to the FMO blur stripe in the image space. Then the latent parts of the original input and the pseudo input are fed into the AffNet to estimate this random affine transformation. Furthermore, a consistency constraint between the transformation in the image space and the transformation in the latent space is also introduced. Please refer to Appendix B for more details. In the finetuning stage, we train the model for 20 epochs in total. For the first 10 epochs, we use the learning rate $lr = 1e - 4$ and set $\alpha_C = 0.01$. The learning rate is reduced to $1e - 5$ and $\alpha_C = 0$ is set for the next 10 epochs. During the training process, the part number of the DIB module is $m = 16$, and the kernel size of the average kernel $K_{avg}$ is $11 \times 11$. In both training stages, we use Adam optimizer (Kingma & Ba, 2015) with batch size 24. The model is trained on 8 NVidia A5000 GPUs and the total training time is about 1.5 days. As for the implementation of the LDINet, the encoder is a variant version of the ResNet50 (He et al., 2016). The shared layers of the decoder are two ResNet bottleneck blocks with up-scaling layers, and the two branches of the decoder are consisting of convolution layers and up-scaling layers. The projector of the DIB module is a ResNet bottleneck block, and the AffNet is composed of four ResNet bottleneck blocks and a predictor consisting of two convolution layers and a ReLU activation. For more implementation details, please refer to Appendix A.

## 3.4 EVALUATION

In this section, we compare the proposed LDINet with the state-of-the-art methods on a variety of datasets. To be specific, we first compare LDINet with the existing FMO deblatting methods based on energy minimization (TbD (Kotera et al., 2019) and TbD-3D (Rozumnyi et al., 2020)) and the data-driven methods (DeFMO (Rozumnyi et al., 2021a) and Bit++(Zhong et al., 2023)). Note that Bit++ predicts the sharp image only, and thus we do not report its trajectory estimation results. We do not compare to SfB (Rozumnyi et al., 2021b) which is based on differentiable rendering, as its performance is highly related to the prior estimation of the silhouettes of the objects and the time consumption is unaffordable. The Peak Signal-to-Noise Ratio (PSNR), Structure Similarity Index Measure (SSIM), and Trajectory Intersection over Union (TIoU) are chosen as the evaluation metrics. Following the protocols from DeFMO (Rozumnyi et al., 2021a), we generate the estimation of the ground truths by averaging over the sequence every 5 frames to match the exposure time of the ground truths in the evaluation datasets. Considering the ambiguity of the direction of motion

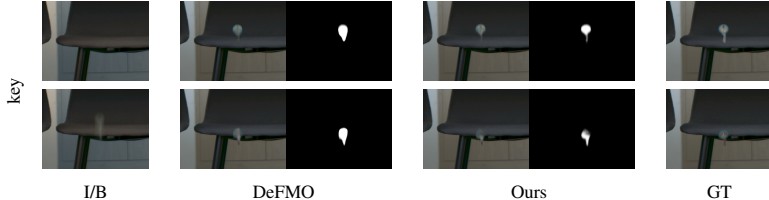

Figure 4: Qualitative comparisons of methods. The leftmost column shows the input pairs, the blurred image $I$ and the background $B$. The rightmost column shows the ground truth. We represent the results for the shapes key from the dataset Falling Objects (Kotera et al., 2020). Our method is compared with the DeFMO (Rozumnyi et al., 2021a). For each method, we show the estimated appearance (left), the estimated mask (right), and the temporal super-resolution frames at $t = 0$ (top) and $t = 1$ (down).

trajectory, we choose the direction with a better PSNR score. The sub-frame trajectory (Figure 3) is estimated using the mass center of the generated estimation of mask $\widehat{M_\tau}$.

The evaluation results are provided in Table 1. The datasets in the table are given in descending order regarding to the shape complexity. It can be observed that the data-driven methods outperform the energy-minimization methods by a wide margin and the performance gap increases as the shapes of the objects in the datasets become more complex. We speculate that this is primarily due to the limitations of the prior assumptions used in the energy-minimization methods. As the objects' shape becomes more complex, these prior assumptions no longer match the distribution of the datasets, resulting in bias errors. This also suggests that the data-driven methods could derive a more precise prior from the training data. Then compared to these existing methods, our LDINet achieves the best performances in most cases on all the three datasets by introducing the decomposition-interpolation block in the latent space. In particular, on the Falling Objects (Kotera et al., 2020), our method outperforms DeFMO by 1.26 dB on the metric PSNR, demonstrating that our method can capture the complex shapes of fast-moving objects. On the TbD-3D and TbD datasets, our method also outperforms DeFMO in most metrics.

## 3.5 ABLATION STUDY

Table 2: **Ablation study: architecture and objectives**. In this table, we do ablation on the structure of the decoder, the introduction of the reversible loss $\mathcal{L}_{id}$, the background reduction loss $\mathcal{L}_L$, the frame consistency loss $\mathcal{L}_C$, and the weighting scheme $W_\tau$. The results on the Falling Object dataset (Kotera et al., 2020) and TbD dataset(Kotera et al., 2019) are listed.

| Arch. | Objective | | | | Falling Objects | | | TbD | | |
|---|---|---|---|---|---|---|---|---|---|---|
| bi-branched | $\mathcal{L}_{id}$ | $\mathcal{L}_L$ | $\mathcal{L}_C$ | $W_\tau$ | TIoU↑ | PSNR↑ | SSIM↑ | TIoU↑ | PSNR↑ | SSIM↑ |
| ✗ | ✓ | ✓ | ✓ | ✓ | 0.689 | 27.39 | 0.755 | 0.590 | 24.74 | 0.598 |
| ✓ | ✗ | ✓ | ✓ | ✓ | 0.686 | 28.08 | 0.771 | 0.605 | 25.07 | 0.615 |
| ✓ | ✓ | ✗ | ✓ | ✓ | 0.687 | 27.66 | 0.767 | 0.606 | 23.79 | 0.577 |
| ✓ | ✓ | ✓ | ✗ | ✓ | 0.681 | 27.50 | 0.763 | 0.605 | 23.80 | 0.581 |
| ✓ | ✓ | ✓ | ✓ | ✗ | 0.679 | 27.34 | 0.753 | 0.612 | 24.44 | 0.592 |
| ✓ | ✓ | ✓ | ✓ | ✓ | 0.692 | 28.09 | 0.770 | 0.611 | 25.31 | 0.617 |

In this section, we conduct ablation studies to analyze the effects of different components and the hyperparameters in the proposed LDINet.

As shown in Table 2, we first observe that the introduction of the bi-branched structure provides significant improvements on the metrics, by separating the estimation of the appearance and the mask. On the other hand, it is seen that reducing the influence of the background on the feature maps with $L_L$ shows a positive impact. However, lacking the regularization term $\mathcal{L}_C$ between the adjacent latent parts that are decomposed in the DIB module results in a significant drop in the performance of the model. Moreover, without the reversible term $\mathcal{L}_{id}$ that keeps the affine transformation in two

directions to be the inverse of each other, the metrics show a slight drop on the Falling Objects dataset but a relatively large drop on the TbD dataset. This indicates that this term would provide some regularization for the prediction of the affine transformations and reduce overfitting on the training set. Finally, the weighting scheme $W_\tau$ also improves the performance of the model by decreasing the supervision strength for the error-prone area (i.e., the border of the objects) and pays more attention to the inner part of the object that are precisely segmented by the estimated mask.

Table 3: **Ablation study: number of latent parts.** This table lists the results on the Falling Object dataset (Kotera et al., 2020) with the number of parts varying in the DIB module.

| Number of parts | Falling Objects | | |
|---|---|---|---|
| | TIoU↑ | PSNR↑ | SSIM↑ |
| 4 | 0.683 | 27.49 | 0.748 |
| 8 | 0.689 | 27.82 | 0.757 |
| 12 | 0.690 | 28.10 | 0.766 |
| 16 | 0.692 | 28.09 | 0.770 |
| 20 | 0.688 | 27.89 | 0.767 |

Table 4: **Ablation study: portion of scalar channels** In this table, we study the effect of the portion of the scalar channels in the latent parts.

| Portion | Falling Objects | | |
|---|---|---|---|
| | TIoU↑ | PSNR↑ | SSIM↑ |
| 1 | 0.684 | 27.76 | 0.757 |
| 1/2 | 0.669 | 27.90 | 0.769 |
| 1/3 | 0.692 | 28.09 | 0.770 |
| 0 | 0.686 | 27.78 | 0.762 |

Next, we investigate the effects of the number of latent parts in the decomposition and the proportion of the scalar channels for the interpolation of the DIB module. The results are provided in Table 3 and Table 4. First, the number of parts in the decompostion of the DIB module controls the fidelity of the DIB module. In particular, with more parts, the time interval between adjacent parts becomes smaller and the transformation between the adjacent parts behaves more likely to a linear transformation, which improves the affine estimation quality. As shown in Table 3, increasing the number of parts in the DIB structure brings a better performance. However, we note that the increase of the number of parts also leads to the explosion of the memory footprint and a heavy calculation burden. Thus, we set the number of parts to 16 in our implementation of the DIB module. On the other hand, the effect of disentangling the latent part into the scalar fields and the gradient fields is shown in Table 4. It is seen that the totally scalar-like (i.e., the portion is 1) or totally gradient-like (i.e., the portion is 0) latent part does not obtain the best performance. This indicates that simply formulating the latent parts as scalar fields or gradient fields is not enough to capture the complex transformation behavior in the latent space under the affine transformation, while the mixture of the scalar fields and gradient fields provides a more appropriate approximation.

### 3.6 QUALITATIVE COMPARISONS

The Qualitative results are given in Figure 3 and Figure 4. From Figure 3, it is seen that DeFMO(Rozumnyi et al., 2021a) fails to deal with the trajectories near the ball's rebounce. However, as illustrated in Figure 4, LDINet generates sharper appearances and more accurate masks. To mention a few, the masks of the falling key generated by LDINet have a higher quality than DeFMO, and the appearances of the falling marker are more precise in our results compared to DeFMO. For more results, please refer to Appendix C.

### 4 CONCLUSION

In this paper, we propose a neural network based deblatting method for deblurring and matting of FMOs. In particular, we introduce a decomposition-interpolation based module in the latent space to incorporate the prior of the temporal sequential structure into the deblatting process and properly generate the target latent frames. Thus the structure of the latent space is fully explored and the feature maps is further disentangled into scalar fields and gradient fields based on the different interpolation behavior under affine transformation. Extensive experiments are conducted and the evaluation results show that our LDINet has achieved superior performances in most cases when compared with the existing methods.

**Reproducibility statement** To facilitate reproducibility, we illustrate the training details in subsection 3.3 and explain the network structure of LDINet in Appendix A. Moreover, we provide our pretraining method in Appendix B.

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

## A THE IMPLEMENTATION OF THE LDINET

### A.1 THE STRUCTURE OF THE ENCODER

The structure of the encoder is based on the ResNet-50 (He et al., 2016) with the nuance that we only take the first three downsampling blocks and extend the last block with five ResNet bottlenecks. The channel number of the feature generated by the encoder is 2048.

### A.2 THE STRUCTURE OF THE DECODER

As shown in Figure 1, the decoder is several shared layers and two convolutional branches. To be specific, the shared layers are two residual blocks. Each residual block is followed by a pixel shuffle layer (Shi et al., 2016) which up-scales the spatial size of the latent frame by a factor of two. The output channel numbers of the residual blocks are 256 and 64, respectively. Given the up-scaled latent frame, we use two convolutional branches to estimate the RGB channels for appearance and the alpha channel for mask respectively. These two branches are similar in structure. In each branch, we first use a $3 \times 3$-conventional layer with the 64 output channels. Then a pixel shuffle layer is applied to up-scale the size of the feature maps by a factor of two. Finally, the feature maps go through two convolutional layers with the numbers of output channels being 16 and 4 respectively and are transformed into outputs with the last layer.

### A.3 The structure of the DIB module

Here we introduce the network structure in the DIB module, including the projector and the AffNet. The projector is a ResNet bottleneck block and its channel number of the output is $512m$ where $m$ is the number of the output parts. The AffNet accepts an input with 1024 channels which is concatenated by two latent parts. The structure of AffNet is shown in 2. The first ResNet bottleneck block reduces the channel number to 64. And the rest three ResNet bottleneck blocks keep the channel number. Finally, the predictor first reduces the channel number from 64 to 16 with the first $3 \times 3$-convolution layer. After a ReLU activation layer, the second $3 \times 3$-convolution layer predicts the 6 parameters for each affine transformation.

## B Pretraining

The prediction of the affine transformations is of significance in the DIB module. It is necessary to train the affine transformation in advance as the prediction of transformations that are not precise enough could impair the training of the model. To guide the training of the AffNet, we propose two objectives, a pseudo supervision loss and a consistency loss.

We illustrate the pretraining stage in Figure 5. Specifically, the channels of the latent part is dispatched into the scalar fields and the gradient fields. Then we generate a pseudo input by applying a small random affine transformation $A$ to the FMO object in the input and align the latent parts of the two inputs in the latent space by applying $A$ to the latent parts of the original input with the application operator $\mathbf{\Phi}$. And we use $A$ as the ground truth to train the AffNet which takes the latent parts with the same time index from the two input images as input. Since the AffNet is not yet accurate enough and the latent space is not well constructed, the interpolation method is used with weighted summation instead in the pretraining stage, as shown in the dashed line part of Figure 5. The weighting scheme $\boldsymbol{v}(\tau)$ in the interpolation is

$$Q_\tau = \sum_i v_{t_i}(\tau) P_{t_i},  \tag{12}$$

where the components $v_{t_i}(\tau) = \frac{\exp(-\sigma(t_i - \tau)^2)}{\sum_{k=1}^m \exp(-\sigma(t_k - \tau)^2)}$ and the time index $t_i = \frac{i-1}{m-1}$. The parameter $\sigma$ is used for adjusting the correlation between the latent frame $Q_\tau$ and the latent parts $P_{t_i}$ and we set $\sigma = 800$ in our implementation. We pretrain the model for 40 epochs with $lr = 1e - 4$ and $\alpha_C = 0.01$.

### B.1 Pseudo supervision for AffNet

Since there is no explicit supervising signal to train the AffNet, we construct a pseudo input with the FMO part transformed by a small affine $A$. And we force the AffNet to estimate the transformation from pairs of latent parts with the same time index from the two different inputs. Here, we denote the latent parts of the original input and the transformed input as $P_i^O$ and $P_i^A$ respectively, and denote the predicted affine transformation from the original pieces to the transformed pieces as $\widehat{A}_i$. Thus the prediction loss is

$$\mathcal{L}_A = \frac{1}{mW_l H_l} \sum_{i=0}^{m-1} \sum_{j=0}^{W_l-1} \sum_{k=0}^{H_l-1} \ell_{MSE}(\widehat{A}_i^{j,k}, A),  \tag{13}$$

where $W_l$ and $H_l$ are the width and height of the latent feature maps, and $\widehat{A}_i^{j,k}$ is the predicted affine transformation on the position $(j, k)$.

### B.2 Consistency between the latent and the image space under affine transformation

Here, we aim to find an appropriate latent space where the features are represented as scalar fields and gradient fields. According to the different behaviors shown by the scalar fields and gradient fields under the affine transformation, we introduce a consistency constraint which forces the transformation results of the latent parts approaching the latent parts which are generated from the inputs

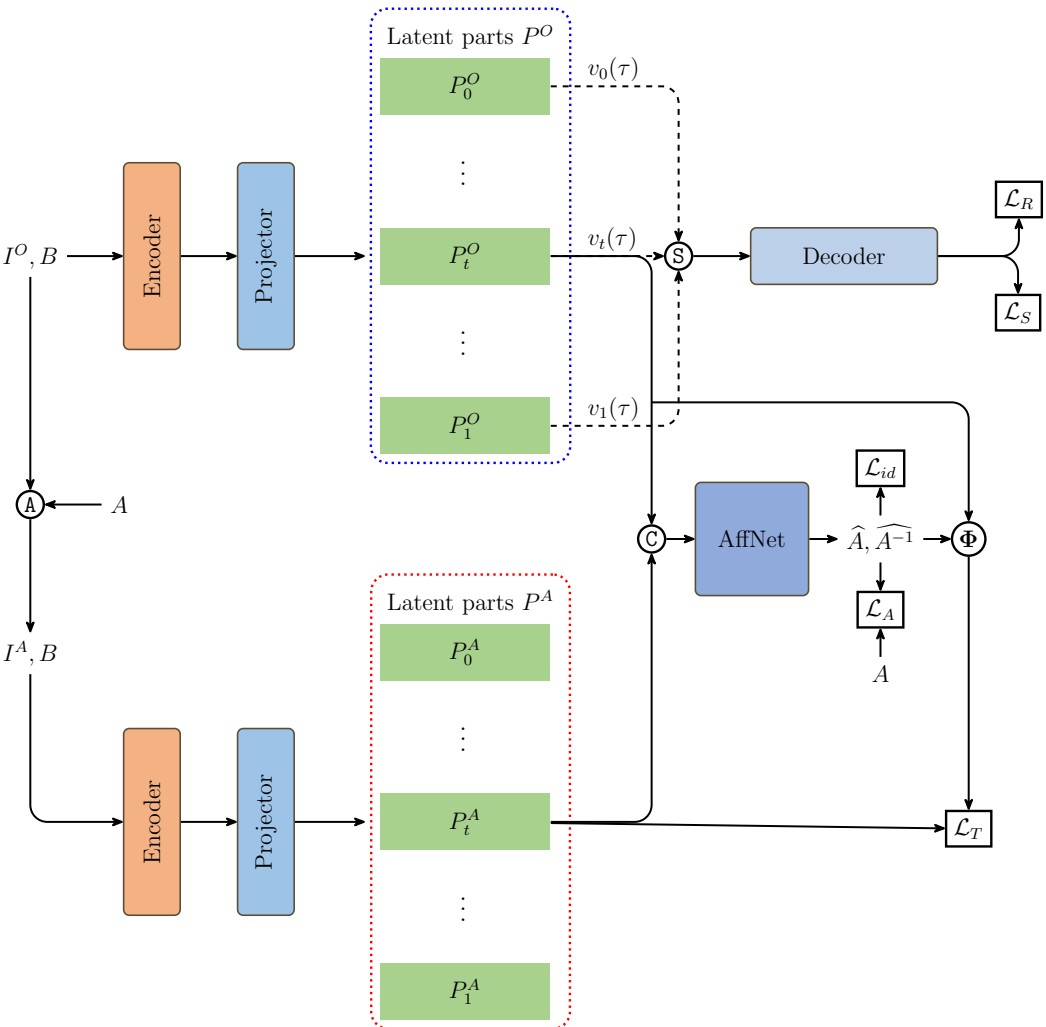

Figure 5: **Pretraining stage**. $I^O, B$ is the original input pair, $I^A, B$ is the pseudo input pair obtained by applying the affine transformation $A$ to $I^A$. The latent parts $P^O$ and $P^A$ are generated from $I^A, B$ and $I^O, B$ respectively. The dashed line part is the surrogate of interpolation in the original pipeline, where we use weighted summation instead of the affine transformation. The AffNet predicts the estimation of the affine transformation $\widehat{A}$ and its inverse $\widehat{A^{-1}}$ for each time index $t \in \{\frac{i-1}{m-1}\}_{i=1}^{i=m}$.

transformed in the image,

$$\mathcal{L}_T = \frac{1}{m} \sum_{i=0}^{m-1} \ell_{MSE}(\mathbf{\Phi}[A, P_i^O], P_i^A) \tag{14}$$

## C   DEBLATTING RESULTS

In this section, we should the extensive deblatting results. Figure 6 shows the results for appearances and masks.

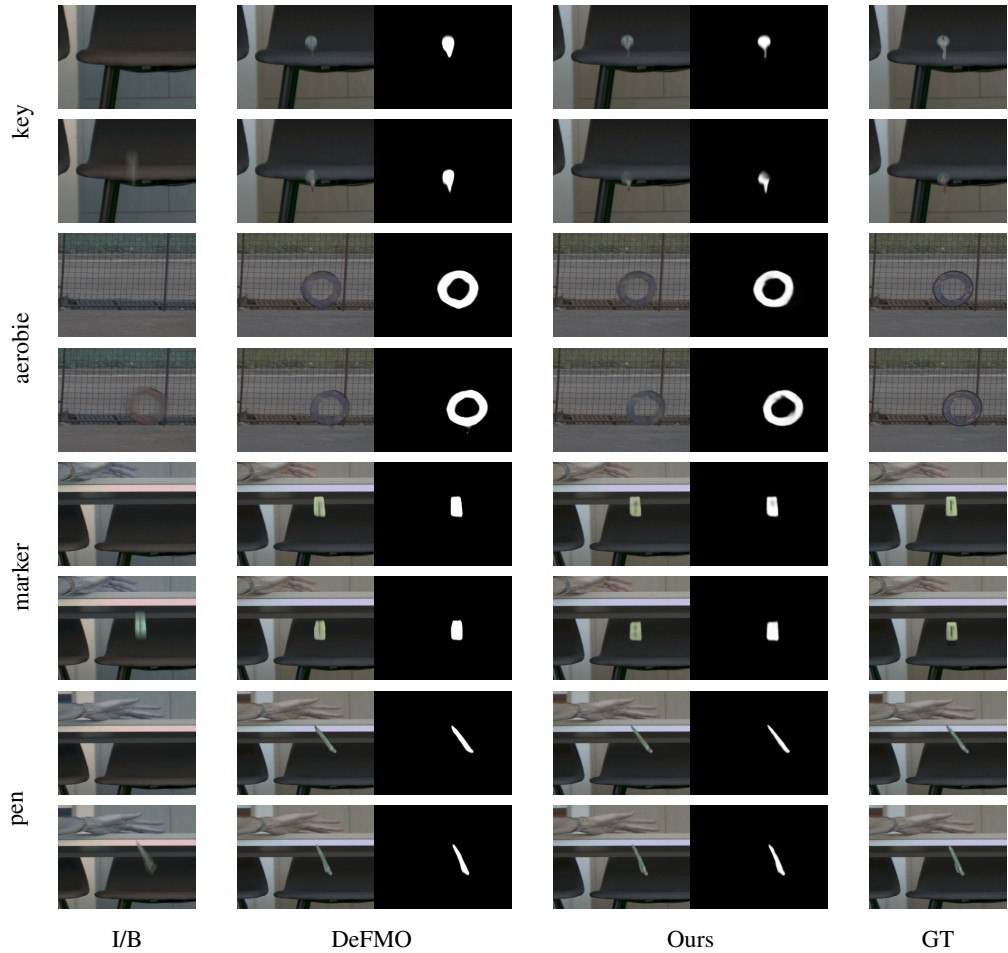

Figure 6: **Extensive deblatting results**. The leftmost column shows the input pairs, the blurred image $I$, and the background $B$. The rightmost column shows the ground truth. We represent the results for the shapes key, marker, and pen from the dataset Falling Objects (Kotera et al., 2020) and aerobie from the dataset TbD-3D(Rozumnyi et al., 2020). Our method is compared with the DeFMO (Rozumnyi et al., 2021a). For each method, we show the estimated appearance (left), the estimated mask (right) and the temporal super-resolution frames at $t = 0$ (top) and $t = 1$ (down).

