# OpenReview forum: "LDINet: Latent Decomposition and Interpolation for Single Image FMO Deblatting"
_ICLR.cc/2024/Conference — Submitted to ICLR 2024_

### Official Review · Reviewer_JWwD · 2023-10-26

**Soundness:** 3 good
**Presentation:** 3 good
**Contribution:** 3 good
**Rating:** 5
**Confidence:** 3

**Summary:**

This paper proposes LDiNet, a novel network for FMO deblatting. The authors propose Decomposition-Interpolation Block (DIB) to break down feature maps into discrete time indexed parts and interpolate them accordingly to obtain the target frame. The authors identify the non-triviality of affine transformation in the latent space. To address this, they decompose each part of the latent representation into a scalar field and a gradient field and train an AffNet to estimate the affine transformation in the feature space. The authors also introduce several objectives to optimize the proposed network. The experimental results demonstrate the superior performance of LDiNet compared to existing methods.

**Strengths:**

1. The idea of interpolation by affine transformation in latent space to diversify the representations of different time indices is interesting.
2. The authors propose a novel technique to handle the non-linearity of affine transformation in the latent space. The observation that the convolution results can be decomposed into scalar fields and gradient fields is non-trivial.
3. The authors provide detailed training settings and comprehensive experimental results, including analysis of the effect of the parameters and some visualization results.

**Weaknesses:**

1. The contributions are somehow incremental. The encoder-decoder framework and most of the training losses mentioned in this article have already been proposed in DeFMO.
2. Although interpolating in hidden space seems to have some intuitive benefits, and the proposed network achieve empirically improvements, the authors do not seem to provide ablation study to verify the improvement of DIB module. The authors should conduct further experiments to explore the impact of the DIB module and add some understanding experiments if possible to illustrate how DIB "exploits the intrinsic structure of the latent space" and thus benefiting the deblatting.
3. I have some doubts about the performance of the proposed method. Though there are only three datasets evaluated in this paper, there are no improvements of PSNR and SSIM on TbD. The improvement of PSNR on TbD-3D also seems marginal. It might be more convincing if the authors ran multiple random seeds and provided the mean and standard deviation or provide results on more datasets.

**Questions:**

1. I'm puzzled as to why the weights in the formula at the bottom of page 4 (which has no formula label) are set this way and why they serve the purpose of "fully leverage the information from both the two neighboring parts."
2. I would like to know the training cost of the proposed method compared to DeFMO if possible.
3. Could you explain the motivation for choosing the direction with a smaller relative error rate when determining the trajectory direction?

---

> ### Author Response · Authors · 2023-11-18
> **Responses for reviewer JWwD**
>
> Thank you for the valuable comments. The following are our responses to the comments.
>
> **Q1: why the weights in the formula at the bottom of page 4 (which has no formula label) are set this way and why they serve the purpose of "fully leverage the information from both the two neighboring parts.**
>
> A1: The weights in this formula are set in such way to guarantee that $Q_{\tau}$ is identity to $P_{\tau}$ when $\tau = t$ or $\tau=t’$. In the case that $\tau \in (t, t’)$, a larger weight is given to the transformed result from the closer latent part and a smaller one to the other transformed result. Since the target latent frame $Q_{\tau}$ is a combination of two transformed results from two neighboring latent parts, it would fully leverage the information from both sides.
>
> **Q2: The training cost of the proposed method compared to DeFMO.**
>
> A2: For the training dataset used in Section 3, it costs about 1.5 days and 2.5 days for the training of our method (60 epochs) and DeFMO (50 epochs) respectively on 8 Nvidia A5000 GPUs.
>
> **Q3: The motivation for choosing the direction with a smaller relative error rate when determining the trajectory direction.**
>
> A3: For the direction selection, we actually follow DeFMO to leverage such a strategy for determining the appropriate trajectory direction. The difference is that our method only calculates the relative error rate on the masks. Since in the generated results, only the masked part of the appearance is what we need, it can be considered that the reconstruction of the mask needs to be focused on.
>
> **Q4: The contributions are somehow incremental. The encoder-decoder framework and most of the training losses mentioned in this article have already been proposed in DeFMO.**
>
> A4: Although our method is built with a similar encoder-decoder framework and loss functions, we want to emphasize that the novelty of our method mainly lies in the decomposition-interpolation paradigm, which allows naturally introducing the required time indexes via interpolation operation. And the well-designed AffNet helps perform feature alignment between different frames during the interpolation. Further, the splitting scheme of the scalar-like and gradient-like feature maps could further improve the alignment effect as the processing way of scalar and gradient fields under the affine coordinate transformations is different. This helps generate a more accurate latent frame at the target time index for decoding. These designs can well distinguish our method from DeFMO. We will add such explanations in the revision.
>
> **Q5: More ablation studies and experiments for the DIB module.**
>
> A5: We have provided an ablation study to compare the performance between settings of linear interpolation and affine interpolation in the DIB module. Please refer to the common response part.
> In addition, we also provide the visualization of their output images in the supplemental material for better understanding.
>
> **Q6: Some doubts about the randomness on the performance of the proposed method.**
>
> A6: Please refer to the common responses part.

---

### Official Review · Reviewer_Gmw4 · 2023-10-30

**Soundness:** 3 good
**Presentation:** 3 good
**Contribution:** 3 good
**Rating:** 8
**Confidence:** 5

**Summary:**

This paper solves a temporal super-resolution task to deblur images of fast-moving rigid objects in static scenes. The method is heavily built on top of the DeFMO method, with several improvements that lead to better qualitative and quantitative scores. Most importantly, the authors encode the input and produce several latent feature maps, which correspond to different time stamps of the underlying motion. Then, these feature maps are interpolated using a new Decomposition-Interpolation Block. Finally, the object is then decoded and rendered at the required time index. The method is trained on a synthetic dataset, but it shows good generalization capabilities since it is evaluated on three real-world datasets. The design choices are evaluated in an extensive ablation study.

**Strengths:**

- The proposed method builds on top of DeFMO and combines in an elegant way this data-driven approach with an idea of piece-wise linear appearance changes from TbD-3D.
- The whole decomposition-interpolation block seems to be well-designed and implemented. The idea of splitting the latent space into time-indexed pieces and then into scalar and gradient fields seems interesting, even though it requires more motivation and analysis.
- The method is extensively evaluated on three real-world datasets. Moreover, many ablation studies are performed, which highlight most of the design choices.
- The paper is mostly well-written and well-structured.

**Weaknesses:**

### **AffNet, scalar and gradient fields**
I do not fully understand why there is a need to split the latent space into scalar and gradient fields. I acknowledge that Table 4 shows that treating everything as scalar (the straightforward case) leads to slightly lower scores. However, I'd like to see an ablation where the feature maps are simply interpolated by linear interpolation. Wouldn't automatic backpropagation (e.g. in PyTorch) solve this? In this case, there is no need for AffNet. P_t's are simply interpolated and become Q_t, without any overhead from AffNet. This is a very important ablation.

On page 4, it says that P''(x) is equal to the gradient of the scalar field S_t, but the scalar field was already defined as P'(x). Is P'(x) = S(x)?

### **Changes w.r.t. DeFMO**
Many parts and loss functions are reused from the DeFMO method. However, some have been modified. For instance, the reconstruction loss compares generated images (I_t) concatenated with the background instead of comparing appearances (F_t), as in DeFMO. Is this change important? Does it bring improvement?

### **Equation (6)**
The sum that denotes the number of pixels within the FMO blur should rather be written as $\sum_D \sum_{\tau} (M_{\tau} > 0)$. Otherwise, I don't understand how it can normalize the losses. Moreover, why is this normalization different for the first (without $>0$) and the second term (with $>0$)?

### **Feature consistency loss**
Feature consistency loss (9) penalizes the differences between adjacent latent feature maps. It means that it's minimized when all latent maps are the same, which shouldn't be the case if the goal is to capture a moving object. This is similar to the time-consistency loss in DeFMO, which at least contains normalized cross-correlation that allows for some movement. In contrast, the feature consistency loss prefers only identical feature maps.

### **Table 2 (arch)**
Arch. (bi-branched) ablation is not clear. The paper says that it provides improvement by separating the estimation of the appearance and the mask. Does it mean that otherwise, they are not separated? What does it even mean? This is very confusing.

In general, I'd like to see more ablations on the architecture side, e.g. does it make sense to introduce AffNet at all? How is it dependent on pre-training?

### **Experimental results**
It's not clear how many times was each ablation/experiment run before reporting the results. Is it run only once? In general, it's always good to run many times and provide mean/std values of each score. For now, it's not clear how much the improvement is contributed by randomness. For example, in Table 3, when the number of parts is set to 20, the scores go down compared to 16. Is it expected?

### **Typos**
There are many typos in the paper:
- Abstract: with backgrounds -> with background
- Abstract: the feature maps of the long blur is -> are
- Abstract: experiments are conducted and has shown -> have shown
- Intro (p1): which is accordance -> is in accordance
- Conclusion: feature maps is -> feature maps are


### **Other comments**
- "Conventional methods mostly recover a clear image at the median of the motion": this is not true, I'd even say they recover a clear image at an arbitrary location of the motion.
- This method is not compared against SfB, which is fine since SfB is used more like a post-processing on top of DeFMO, meaning that the proposed method can actually be used to get better results from SfB. However, it would be really nice to see SfB results if this new method is used for silhouette estimation. I believe SfB would perform even better.

**Questions:**

- Are AffNet and separating gradient fields really necessary?
- It would be helpful to add SfB results if sub-frame silhouettes are used from the proposed method as input.
- Please see the Weaknesses section.

---

> ### Author Response · Authors · 2023-11-18
> **Responses for reviewer Gmw4**
>
> Thank you for the valuable comments. The following are our responses to the comments.
>
> **Q1: Are AffNet and separating gradient fields really necessary?**
>
> A1: Please refer to the common responses part.
>
> **Q2: It would be helpful to add SfB results if sub-frame silhouettes are used from the proposed method as input.**
>
> A2: We have provided the results for SfB with silhouettes generated by LDINet as follows:
> | Method |Falling TIoU| Falling PSNR|Falling SSIM| TbD-3D TIoU | TbD-3D PSNR | TbD-3D SSIM | TbD TIoU |TbD PSNR| TbD SSIM |
> | :-: | :-: | :-: | :-: | :-: |:-: |:-: |:-: |:-: |:-: |
> |  SfB (ours)  | 0.658(0.012) | 25.348(0.421) | 0.730(0.004) | 0.843(0.003) | 24.172(0.096) | 0.651(0.001) | 0.601(0.000) | 24.130(0.062) | 0.609(0.001) |
> |SfB (DeFMO)| 0.650(0.001) | 24.982(0.096) | 0.732(0.003) | 0.864(0.000) | 23.982(0.027) | 0.640(0.002) | 0.561(0.001) | 24.164(0.030) | 0.592(0.002) |
>
> Since time is limited, we only report the 8-step results with two prototypes and set the learning rate to 0.03 for stable optimization in SfB. And the std is provided in parentheses for 3 times experiments. We will add this comparison into the revision.
>
> **Q3: An ablation where the feature maps are simply interpolated by linear interpolation.**
>
> A3: Please refer to the common response part.
>
> **Q4: On page 4, it says that $P’’(x)$ is equal to the gradient of the scalar field $S(x)$, but the scalar field was already defined as $P’(x)$. Is $P’(x) = S(x)$?**
>
> A4: $P’(x)$ is not $S(x)$. Due to the separating of scalar-like and gradient-like feature maps, $P’’(x)$ and $P’(x)$ are generated from different channels of the same latent part. Thus, though $S(x)$ is the scalar field to produce $P’’(x)$, there actually exists differences between $P’(x)$ and $S(x)$.
>
> **Q5: Loss changes w.r.t. DeFMO.**
>
> A5: Please refer to the common responses part.
>
> **Q6: For the Equation 6, (1) about the sum term that denotes the number of pixels within the FMO blur, and (2) why is this normalization different for the first (without $>0$) and the second term (with $>0$)?**
>
> A6: Since $I$ denotes the blur image which is the average of a series of sharp images at the given time indexes within the exposure time, the formula reviewer provided could count the same pixel multiple times because of its appearance in masks of different time indexes. In our formula, we use a pixel-wise indicator function $[\cdot >0]$ to set each pixel that appears in any mask to 1 which avoids the multiple-counting.
>
> On the other hand, $>0$ is part of the pixel-wise indicator function $[\cdot >0]$. The first normalization term is without the function because the mask $M_\tau$ is a binary map whose value is either 0 or 1 for different pixels and thus it could be directly summed. While the second normalization needs the function to convert the summation over time indexes $\sum_{\tau} M_{\tau}$ to a binary map to further count the size of the blurred area.
>
> **Q7: The Arch. (bi-branched) ablation in Table 2 and the dependence of pretraining.**
>
> A7: In the decoder, we separate the estimation of the appearance and the mask with two different branches as in Figure 1. In particular, this separation happens after the first two Resblocks of the decoder. Otherwise, they are estimated with one shared branch setup. We will add more explanations to the revision.
> In addition, for the dependence of pretraining, we provide the following ablation study to show its advantage:
> | Method |Falling TIoU| Falling PSNR|Falling SSIM| TbD-3D TIoU | TbD-3D PSNR | TbD-3D SSIM | TbD TIoU |TbD PSNR| TbD SSIM |
> | :-: | :-: | :-: | :-: | :-: |:-: |:-: |:-: |:-: |:-: |
> |DeFMO | 0.684| 26.83 |0.753 |0.879 |26.23| 0.699 | 0.550 | 25.57 | 0.602|
> |w/ pretrain| 0.686(0.007) | 28.087(0.006) | 0.771 (0.001) | 0.906(0.002) | 26.503(0.136) | 0.707(0.003) | 0.616(0.004) | 25.242(0.122) | 0.626(0.008) |
> | w/o pretrain| 0.697 |27.706 | 0.766 | 0.906 | 26.490 | 0.706 | 0.615 | 25.702 | 0.626 |
>
> Since time is limited, we only report the one-run results for w/o pretraining here and we will further complement the results.
>
> **Q8: About the randomness of the experimental results.**
>
> A8: Please refer to the common responses part.
>
> **Q9: About the trend change in Table 3.**
>
> A9: For “In Table 3, it is seen that the score goes down if the number of parts is set to 20.”, this is expected as the final results would rely more on the quality of the direct output of the encoder when the number of parts increases and the affine transformation would not produce effects as the space for interpolation is gradually reduced.
>
> **Q10: About the typos and some expressions.**
>
> A10: Thanks a lot, and we will correct the typos and revise our paper more carefully. For the description of "Conventional methods mostly recover a clear image at the median of the motion", such expression is only to illustrate the common problem setup in the existing literatures. We will revise accordingly to avoid confusion.

---

### Official Review · Reviewer_3qEi · 2023-10-31

**Soundness:** 3 good
**Presentation:** 2 fair
**Contribution:** 3 good
**Rating:** 6
**Confidence:** 3

**Summary:**

This paper tackles the problem of FMO deblurring. The goal of this paper is to separate the object from the background and recover the appearance of the foreground object for each time stamp during the exposure time. In particular, the proposed method decomposes the latent space into a few latent segments. A pair of latents could be related by the predicted affine transformations from a pair of time stamps. And the latent at any time stamp could be interpolated from the disentangled latents and it could be decoded to reconstruct the mask and image at time t. The method is evaluated on FMO deblatting benchmark which shows promising results, outperforming existing methods on two out of three existing datasets.

**Strengths:**

+ The idea is promising in recovering the appearance of the FMO and has potential to reconstruct the sharp image of FMO at any time during the exposure time.
+ The decomposition of the latent into a set of latent segments which are responsible different latent frames. The interpolation network could lead to the reconstruction of the latent at any time within the exposure time.

**Weaknesses:**

1)	Convolution is defined as a linear combination of the signals within a spatial window in an image. However, it is not clear to the reviewer why we need to further decompose the convolution as a linear combination of summation and series of directional derivatives. What is the benefit of this further decomposition? It seems that the Taylor expansion of the latent is applied at time t to approximate the latent at any time t? It was not clearly motivated in the paper.
2)	Affine transformation. The paper mainly describes what has been implemented to approximate the affine transformation via prediction of two affine transformations (one is computed from I_t to I_t’ and the other one is computed from I_t’ to I_t). Would this be redundant as one transformation should be the inverse of the other?
3)	Eq.10 is not correct. It should be the Frobenius Norm for computing the relative distance between two matrices not the standard MSE.
4)	There is no guarantee that the predicted affine transformation is invertible. While losses are introduced to guide the learning process, no hard constraint is enforced in the framework.
5)	All the equations should be written properly. D is referred to as the domain for the pixel coordinate. However, no index ever appears in the loss function for Eq. (5,6,7,8).
6)	No supplementary is provided in the submission. Could the authors show the recovery of the appearance of the FMO in a video as the proposed method can reconstruct the image at any time within the exposure time? The qualitative results in Figure 6 in the appendix cannot demonstrate the effectiveness of the proposed method. The masks recovered by the proposed method and DeFMO look the same.

**Questions:**

-Please addressed the concerns mentioned above in the weakness section.
Overall, the idea is promising. The only concern from the reviewers is about the motivation of representing convolution as the scalar and directional derivative field. The equations could be improved. The quantitative results are not impressive. It would be great to show more qualitative results.

---

> ### Author Response · Authors · 2023-11-18
> **Responses for reviewer 3qEi**
>
> Thank you for the valuable comments. The following are our responses to the comments.
>
> **Q1: Why we need to further decompose the convolution as a linear combination of summation and series of directional derivatives.**
>
> A1: The main reason is to help improve the alignment effect by the affine transformation. In particular, by decomposing the convolution as a linear combination of summation and series of directional derivatives, the resulting feature maps are correspondingly split into scalar-like and gradient-like feature maps. Further, the scalar-like and gradient-like feature maps have different processing ways under the affine coordinate transformation. Thus, different ways of transformation operations are enforced on these two kinds of feature maps to fully exploit their particular characteristics, and thus better alignment effect could be achieved, which helps improve the embedding quality for any time index and obtain better overall performance.
>
> **Q2: About the affine transformation. The paper mainly describes what has been implemented to approximate the affine transformation via prediction of two affine transformations. Would this be redundant as one transformation should be the inverse of the other?**
>
> A2: We predict the two affine transformations mainly for the consideration of the numerical stability of the model. Empirically, using the inverse operation directly has the risk of numerical instability, especially in the first several epochs in the training. Such a design of modeling the transformation from both directions also provides some flexibility when either one happens to be not precise.
>
> **Q3: The Equation 10 is not correct.**
>
> A3: Thanks for pointing this out. We will correct it by using the Frobenius Norm for computing the relative distance between two matrices.
>
> **Q4: There is no guarantee that the predicted affine transformation is invertible.**
>
> A4: For the invertibility, based on the experimental statistics, the id loss in Equation 10 is less than 1e-4 during the training process, which empirically shows that the predicted affine transformation approximates to be invertible. We will add such analysis in the revision.
>
> **Q5: All the equations should be written properly.**
>
> A5: Thanks for pointing them out. We will revise the Equations 5, 6, 7, and 8 accordingly.
>
> **Q6: About the qualitative results.**
>
> A6: In the supplemental material, we have provided the compared results for the recovery of the appearance of the FMO in the form of videos.

---

### Official Review · Reviewer_wbgD · 2023-11-01

**Soundness:** 2 fair
**Presentation:** 1 poor
**Contribution:** 1 poor
**Rating:** 3
**Confidence:** 5

**Summary:**

This paper introduces a network to interpolate deblurred images with respective object alpha matte for single image with fast moving object. In the proposed network, the features are affine transformed to generate the features in arbitrary time steps. Also, 'scalar fields' and 'gradient fields' are considered in the network. The provided experimental results show the proposed method performs better than other methods. This paper is poorly written and should not be accepted in its current form.

**Strengths:**

The provided experimental results show the proposed method performs better than other methods.

**Weaknesses:**

1. The novelty is quite limited.
2. The authors fail to provide evidence of why the proposed method works. Also, the presentation of this paper is too poor to be fully understood. For example:
(1) More details should be added to discuss the difference between the proposed network and the prior work DeFMO and describe why the proposed method is better.
(2) What are scalar-like and gradient-like feature maps? How does the network get them? Why are they so important? It also lacks an ablation study to validate it.
(3) The detailed process from eq. 3 to eq. 5 is not clear which makes it hard to understand. Also, how this process is presented in Fig. 2 is also not clear.

**Questions:**

See weaknesses for details.

---

> ### Author Response · Authors · 2023-11-18
> **Responses for reviewer wbgD**
>
> Thank you for the valuable comments. The following are our responses to the comments.
>
> **Q1: The novelty is limited.**
>
> A1: Please refer to the common responses part.
>
> **Q2: Fail to provide evidence of why the proposed method works.**
>
> A2: Thanks for your suggestion. We would reorganize the paper and provide more required explanations for the important components of the proposed method as is pointed out. To further validate the effectiveness of our work, we would add more ablation/experimental studies. For example, (1) We would compare with the results by linear interpolation to show the benefits of the affine transformation in our interpolation method, (2) We would provide an ablation setting to show the importance of the pre-training, and (3) We would provide more qualitative results to show the benefits of the decomposition-interpolation paradigm.
>
> **Q3: The presentation of this paper is too poor to be fully understood. (1) More details about the difference between the proposed network and the prior work DeFMO. (2) More details about the implementations for scalar-like and gradient-like feature maps. (3) More details for the process from eq. 3 to eq. 5.**
>
> A3: We will revise the paper accordingly.
>
> (1)	The main difference between our method and DeFMO lies in the way of constructing latent embedding for different time indexes. DeFMO concatenates the target time index with shared latent feature maps as the corresponding embedding for the subsequent decoding. Instead, our method first decomposes the feature maps in latent space into several latent parts and then generates the required corresponding embedding for any time index via an interpolation method with affine transformation. This helps greatly improve the flexibility of generating diverse outputs for different time indexes. Further, in the decoder part, DeFMO generates the mask and appearance of the object with only one branch, while our method generates them with two separate branches as they have different value distributions. More details about the difference will be added in the revision.
>
> (2) For the scalar-like and gradient-like feature maps, such design mainly considers the characteristics of convolution operation under the affine coordinate transformations to improve the interpolation quality. In particular, on the one hand, the convolution operation could be seen as a linear combination of summation and series of directional derivatives, where the directional derivatives are linear projections of the gradient fields. On the other hand, the processing way of scalar and gradient fields under the affine coordinate transformations is different. Thus, it is very beneficial to represent the feature maps output by the convolution operation as a combination of scalar-like and gradient-like feature maps. Such a well-designed splitting scheme helps improve the alignment effect by the affine transformation.
> In practice, our model learns the scalar-like and gradient-like feature maps with a pre-training stage. In the pre-training, we first obtain the latent parts for the original input image. Then we split the channels of latent parts into the scalar-like and gradient-like categories where each scalar-like feature map has a single channel and each gradient-like feature map has two channels as the two components of the gradient field, which are processed by an estimated affine transformation by AffNet. Further, the resulting feature maps are forced to be consistent with the corresponding ones for a distorted input version which is produced by applying a pre-set affine transformation to the FMO blur contained in the original input image. Meanwhile, in this process, the estimated affine transformation is also encouraged to be similar to the pre-set one. The related ablation study is provided in Table 4 to validate the importance of generating two kinds of feature maps instead of only one kind. More details will be added in the revision accordingly.
>
> (3) In particular, Equation 3 and Equation 4 formulate the supposed transformations for the scalar-like and gradient-like feature maps, respectively. In Equation at the bottom of page 4, a function $\Phi$ is first defined to split the input feature map into the scalar-like and gradient-like feature maps, which are then processed with the transformations in Equation 3 and 4 correspondingly. The two respective results are concatenated as the output. Therefore, the Equation 3 and 4 provide the important component for the operation of the Equation at the bottom of page 4. For Figure 2, since it is defined that the latent feature map $P_t$ is the concatenation of the scalar part $P_t’$ and the gradient part $P_t’’$, only the process of the Equation at the bottom of page 4 is explicitly presented in Figure 2(a), where the function $\Phi$ takes as input ($P_t$, $A_{\tau\rightarrow t}$) and ($P_{t’}$, $A_{\tau\rightarrow t’}$) for generating $Q_{\tau}$. More explanations will be added in the revision.

---

### Author Response · Authors · 2023-11-18
**Common responses to reviewers**

We thank all the reviewers for their efforts and valuable comments. Here are the responses to the common concerns for the paper.

**Q1: About the novelty.**

A1: We want to emphasize that the novelty of our method mainly lies in two aspects. First, we propose a new decomposition-interpolation paradigm for the latent space learning of the FMO deblatting task. It can well exploit the intrinsic structure of the latent space and allow naturally introducing the required time indexes via interpolation operation. Meanwhile, the estimated affine transformation by the well-designed AffNet could perform effective alignment between the frames during the interpolation. This helps enable more potentially complex motions to be rendered by the decoder. Second, for the decomposition and interpolation process, we propose to split the latent feature map into a scalar part and a gradient part, which are processed in two different ways under the affine coordinate transformations. This further improves the alignment effect and thus produces more accurate latent frames at the target time indexes for decoding. We will add more explanations in the revision.

**Q2: About the necessity of the AffNet and the scheme of separating scalar and gradient fields.**

A2: For AffNet, it helps perform effective feature alignment with the estimated affine transformations during the interpolation process, which is very important as relative motions usually exist between the output frames for the FMO deblatting task. Such ability is not available for a simple linear interpolation. Further, separating gradient fields is also necessary for releasing the full potential of interpolation via affine transformation. It is based on two aspects: 1) convolution operation could be seen as a linear combination of summation and a series of directional derivatives, where the directional derivatives are linear projections of the gradient fields; 2) the processing way of scalar and gradient fields under the affine coordinate transformations is different. Thus, this separating scheme helps exploit these properties to achieve a better alignment effect for the generation of intermediate embedding for any time index. We will add such explanations in the revision.

**Q3: About the randomness of the experimental results.**

A3: We have run the main experiment 3 times and the results are reported in the table below. (The std is provided in parentheses.) These results correspond to those of Table 1 in the paper. And the DeFMO results are reported based on its original paper.
| Method | Falling TIoU| Falling PSNR| Falling SSIM| TbD-3D TIoU | TbD-3D PSNR | TbD-3D SSIM | TbD TIoU | TbD PSNR| TbD SSIM |
| :-: | :-: | :-: | :-: | :-: | :-: | :-: | :-: | :-: | :-: |
|DeFMO | 0.684| 26.83 |0.753 |0.879 |26.23| 0.699 | 0.550 | 25.57 | 0.602|
|LDINet| 0.686(0.007) | 28.087(0.006) | 0.771(0.001) | 0.906(0.002) | 26.503(0.136) | 0.707(0.003) | 0.616(0.004) | 25.242(0.122) | 0.626(0.008) |

**Q4: About the ablation between linear interpolation and AffNet.**

A4: We have conducted the ablation study where the feature maps are simply interpolated by linear interpolation and the results are provided in the table below. It is shown that the simple linear interpolation performs worse than our method with AffNet, which validates the effectiveness of our design. We have run 3 times for this comparison and the std is provided in parentheses. And the DeFMO results are reported based on its original paper.
| Method | Falling TIoU| Falling PSNR| Falling SSIM| TbD-3D TIoU | TbD-3D PSNR | TbD-3D SSIM | TbD TIoU |TbD PSNR| TbD SSIM |
| :-: | :-: | :-: | :-: | :-: |:-: |:-: |:-: |:-: |:-: |
|DeFMO | 0.684| 26.83 |0.753 |0.879 |26.23| 0.699 | 0.550 | 25.57 | 0.602|
|Linear Interp.| 0.681(0.001) | 27.644(0.137) | 0.762(0.003) | 0.908(0.008) | 26.395(0.060) | 0.705(0.002) | 0.603(0.005) | 25.202(0.044) | 0.615(0.005) |
|Affine Interp.| 0.686(0.007) | 28.087(0.006) | 0.771(0.001) | 0.906(0.002) | 26.503(0.136) | 0.707(0.003) | 0.616(0.004) | 25.242(0.122) | 0.626(0.008) |

**Q5: About loss changes w.r.t. DeFMO.**

A5: The main change of the loss functions lies in the usage of binary cross entropy loss instead of $L1$ loss for the reconstruction of the mask. Further, the introduction of normalization terms, like $\sum_D M_\tau$, is important for stable optimization. The change for the reconstruction loss using generated images $I_t$ concatenated with the background can bring some smoothness and has slight improvement for the results.

For the feature consistency loss, we agree that in the worst case, our proposed feature consistency loss would produce identity feature maps. However, in practice, the weight for this loss is relatively small and helps obtain smoothly changed feature maps. The reason for dropping normalized cross-correlation loss is that the computation cost is relatively large for our optimization setup. More explanations will be added in the revision.

---

> ### Comment · Area_Chair_8ZTs · 2023-11-19
> **Author-Reviewer Discussions Fri, Nov 10 – Wed, Nov 22**
>
> Dear reviewers,
> Could you please read the authors' responses and give your feed back? The period of Author-Reviewer Discussions is Fri, Nov 10 – Wed, Nov 22.
> Many thanks,
> AC

---

### Meta-Review · Area_Chair_8ZTs · 2023-12-06

**Metareview:**

This paper introduces an efficient decomposition-interpolation module (DIB) to generate appearances and shapes of objects for deblurring stripes and separating objects from backgrounds in a single image. Experiments show the effectiveness of the proposed method.

The paper received one “reject” rating, one “marginally below the acceptance threshold” rating, one “marginally above the acceptance threshold” rating, and one “accept” rating.

Both Reviewer wbgD and Reviewer JWwD don’t think the contributions are sufficient. All reviewers think some important parts are not clear, in particular, the differences between the proposed method and DeFMO.

After rebuttals, Reviewer wbgD still thinks the novelty is limited and the improvement is insignificant. Reviewer wbgD still keeps the rating unchanged.

Besides，Reviewer JWwD thinks more ablation studies are needed and has some doubts about the performance of the proposed method. After rebuttals, the Reviewer JWwD decreased the rating from “6” to “5”.

Reviewer wbgD thinks the presentation has problems, including methods, equations, experiments, tables, and writings. In particular, the reviewer thinks many parts and loss functions are reused from the DeFMO method. Some are changed. But it is not clear whether the changes are important or bring improvements. After rebuttals, Reviewer wbgD agreed that the novelty w.r.t. DeFMO is slightly limited and the paper is thought as a borderline paper.

Based on the above comments, the paper still needs more revisions and the decision was to reject the paper.

**Justification For Why Not Higher Score:**

Both Reviewer wbgD and Reviewer JWwD don’t think the contributions are sufficient. All reviewers think some important parts are not clear, in particular, the differences between the proposed method and DeFMO.

After rebuttals, Reviewer wbgD still thinks the novelty is limited and the improvement is insignificant. Reviewer wbgD still keeps the rating unchanged.

Besides，Reviewer JWwD thinks more ablation studies are needed and has some doubts about the performance of the proposed method. After rebuttals, the Reviewer JWwD decreased the rating from “6” to “5”.

Reviewer wbgD thinks the presentation has problems, including methods, equations, experiments, tables, and writings. In particular, the reviewer thinks many parts and loss functions are reused from the DeFMO method. Some are changed. But it is not clear whether the changes are important or bring improvements. After rebuttals, Reviewer wbgD agreed that the novelty w.r.t. DeFMO is slightly limited and the paper is thought as a borderline paper.

The paper still needs more revisions.

**Justification For Why Not Lower Score:**

N/A

---

### Decision · Program_Chairs · 2024-01-16

Reject